# Mitochondrial Ferritin: Its Role in Physiological and Pathological Conditions

**DOI:** 10.3390/cells10081969

**Published:** 2021-08-03

**Authors:** Sonia Levi, Maddalena Ripamonti, Marko Dardi, Anna Cozzi, Paolo Santambrogio

**Affiliations:** 1School of Medicine, Vita-Salute San Raffaele University, Via Olgettina 58, 20132 Milan, Italy; ripamonti.maddalena@hsr.it (M.R.); m.dardi@studenti.unisr.it (M.D.); 2Division of Neuroscience, San Raffaele Scientific Institute, 20132 Milan, Italy; cozzi.anna@hsr.it (A.C.); santambrogio.paolo@hsr.it (P.S.)

**Keywords:** iron, mitochondrial iron metabolism, mitochondrial ferritin, cytosolic ferritins, iron-related disorders

## Abstract

In 2001, a new type of human ferritin was identified by searching for homologous sequences to H-ferritin in the human genome. After the demonstration that this ferritin is located specifically in the mitochondrion, it was called mitochondrial ferritin. Studies on the properties of this new type of ferritin have been limited by its very high homology with the cytosolic H-ferritin, which is expressed at higher levels in cells. This great similarity made it difficult to obtain specific antibodies against the mitochondrial ferritin devoid of cross-reactivity with cytosolic ferritin. Thus, the knowledge of the physiological role of mitochondrial ferritin is still incomplete despite 20 years of research. In this review, we summarize the literature on mitochondrial ferritin expression regulation and its physical and biochemical properties, with particular attention paid to the differences with cytosolic ferritin and its role in physiological condition. Until now, there has been no evidence that the alteration of the mitochondrial ferritin gene is causative of any disorder; however, the identified association of the mitochondrial ferritin with some disorders is discussed.

## 1. Introduction

Iron is essential for many biological functions; however, it is also potentially toxic for its capacity to catalyze free radical formation via the Fenton reaction. Consequently, its homeostasis is strictly regulated at both systemic and cellular levels by complex mechanisms, reviewed in greater detail in [1]. Iron trafficking in and out of the cell is mainly driven by endocytosis of the complex transferrin/transferrin receptor for uptake, and the ferroportin/hepcidin complex for export [2]. Once in the cell, iron is mostly utilized in mitochondria as the cofactor of enzymes involved in energy production. A detailed description of mitochondrial iron metabolism is covered in other recent reviews [2,3]. Concisely, iron enters the outer mitochondrial membrane via a still-unknown chaperone or by direct contact with endosomes, a pathway called kiss-and-run, or by a mitochondrial form of divalent metal transporter 1 [4]. Iron transport across the inner mitochondrial membrane occurs via the mitoferrins (MFRN1 and 2) [2]. Inside the mitochondria, iron is utilized to synthesize the heme and iron-sulfur cluster (ISC), cofactors that are used by the mitochondria itself for sustaining the respiratory chain and TCA cycle or exported by the heme-carrier feline leukemia virus subgroup C receptor 1 (FVLCR1b) and ISC-carrier ATP binding cassette subfamily B member 7 (ABCB7) [2] for the activity of cytosolic enzymes. The excess mitochondrial iron can be stored in mitochondrial ferritin (FtMt). 

Ferritin is an iron storage protein and one of the most important proteins involved in iron metabolism. Cytosolic ferritin is basically present in all tissue cells. Its biochemical and structural properties make ferritin necessary for cell vitality; it is very well preserved in various organisms, from bacteria to humans [5]. The reaction between iron and ferritin results in the formation of the iron core with hydrogen peroxide (H_2_O_2_) as a by-product. The ferroxidase activity, present on the ferritin H-chain, plays a central role in regulating iron availability [6]. This catalytic activity affects various processes, such as cell proliferation and resistance to oxidative damage and consumes the same reagents that are used in the Fenton reaction [7]. Since the Fenton reaction is toxic to the cell, ferritins play an antioxidant role and control the availability of the ferrous ion Fe^2+^ inside the cell [7]. For this reason, all ferritins are organized in a hollow spherical structure where redox reactions take place. The ability to form a compact nucleus of iron separated from the cytoplasm, keep it soluble, and make it biologically available when needed, is typical of all ferritins.

In humans, there are three functional genes for ferritins: *FTH1* on human chromosome 11 which encodes the cytosolic heavy chain (H-chain, FtH, 183 amino acids long), *FTL* on chromosome 19 coding for the light chain (L-chain, FtL, 175 amino acids long), and the intron-free gene *FTMT* located on chromosome 5 coding for the mitochondrial ferritin (FtMt) precursor composed of 242 amino acids (Table 1) [8]. The H- and L-chains are expressed in the cytosol compartment and are under the same iron-dependent post-transcriptional control, but do not tend to assemble into homopolymer molecules. They are predisposed to co-assemble in different proportions into 24-subunit heteropolymers giving rise to a large number of isoferritins with a specific tissue distribution [9]. Instead, mitochondrial ferritin subunit precursors are conveyed to the mitochondria, where the specific N-terminal driving sequence is cut. After the cleavage of the leader peptide, the mature subunits assemble into a 24-subunit homopolymer molecule (FtMt) [10]. Despite this difference, FtMt has functional and structural properties similar to cytosolic ferritin, such as storage and detoxification of cellular iron. Unlike cytosolic ferritin, which is ubiquitously expressed in mammalian tissues, the expression of mitochondrial ferritin is restricted to specific cell types [11]. Analysis of in vitro and in vivo models indicated that mitochondrial ferritin plays a role in protecting mitochondria from damage induced by iron excess. Thus, the study of this protein has deepened the knowledge of mitochondrial iron metabolism [12] and can also be important in the diagnosis of various diseases.

## 2. FtMt: Structural, Biochemical, Functional, and Regulatory Peculiarities

The presence of a specific protein that serves to deposit iron in mitochondria was confirmed only at the beginning of the 21st century thanks to the availability of information about the human genome. The *FTMT* was identified in humans on chromosome 5q23.1 because of its similarity to the cDNA coding sequence of human FtH [10]. Homologous genes have been identified in mammals such as mice, rats, dogs, and chimpanzees [8]. Plants and *Drosophila melanogaster* possess the *FTMT* gene but it is less similar to mammals’ genes, as it is interspersed with introns indicating a different phylogenetic origin [13,14]. The human mRNA of FtMt is about 1 kb long, similar to the transcribed ferritins H and L, but does not contain the typical IRE consensus sequence [8] (Table 1). 

The amino acid sequence of mammalian mitochondrial ferritin is highly conserved and is characterized by an extension of the N-terminal of 60 residues, which represents a signal sequence for transport in the mitochondria and the presence of seven amino acids necessary for ferroxidase activity. Experiments with mitochondrial ferritin overexpression in HeLa cells indicated that the human precursor peptide (MWa 30 KDa), once imported into the mitochondria, is cut and shortened by 58 amino acids [10]. The mature peptide (22KDa) assembles into the 24-subunit structure only when it is within the mitochondrial matrix and no free subunits are revealed by specific anti-FtMt antibodies in any cyto/histochemical analysis (personal communication). In humans, the mature protein, after processing the N-terminal guide sequence, shows 79% identity with ferritin H and 63% identity with ferritin L [8]. Crystallographic analysis of recombinant proteins produced in *Escherichia coli* has shown that the three-dimensional structures of FtMt and FtH are very similar, but some differences have been observed in the localization and presence of iron-binding sites [15]. The protein folds into five α-helices identified as A, B, C, D, and E (Figure 1). Most of the substitutions between rHuHF and rFtMt occur on the outer surface, except for the substitutiona Y12F, I133L, and A144S [15]. This last substitution reduces the effectiveness of the FtMt ferroxidase center. Indeed, kinetic analyses of oxygen consumption and iron binding revealed that, although they have a similar ferroxidase center, the two proteins, FtMt and FtH, have different iron oxidation and hydrolysis chemistry [16]. The ferroxidase activity of FtMt is slower than that of ferritin H. This property and the fact that FtMt is a homopolymer could be related to the particular environment of the mitochondria, where iron is most available, and the concentration of reactive oxygen species (ROS) is high. In any case, the homopolymer is functional in incorporating mitochondrial iron, and the expression of FtMt has profound effects on the cellular homeostasis of iron itself. The presence of high amounts of FtMt causes iron to relocate from the cytosol to the mitochondria [17,18]. This, in turn, induces a high accumulation of iron within the mitochondria and, consequently, a reduction in the availability of cytosolic iron [17,18]. 

Northern blot analysis in human samples showed that the FtMt coding transcript is present at high levels in the testes, while RT-PCR analyses identified smaller amounts of mRNA in the brain, kidneys, and thymus [8]. Unlike the cytosolic ferritins, the transcript was not identified in iron-storage tissues such as the liver and spleen. FtMt expression is also high in normal spermatozoa [19] and in sideroblasts from patients with sideroblastic anemia [20]. A detailed immuno-histochemical study of murine tissues showed the presence of FtMt protein in the testis, heart, spinal cord, kidneys, pancreatic islets of Langherans cells, and smooth muscle tissue [11]. All types expressing FtMt were also characterized by high metabolic activity. An example is Leydig cells of the testis, which are characterized by high energy needs, because of their function in hormone biosynthesis; the expression of FtMt in these cells was found to be very high. Among germ cells, the ones showing the highest level of FtMt were mature sperm cells, where mitochondria respond very actively to the high energy needed for movement. In the normal retina, FtMt showed a detailed localization in the inner segments of photoreceptors, a cellular specialization known to express a high number of mitochondria; moreover, widespread distribution of FtMt was found in the inner layer of the retina. The expression of FtMt in these regions, on the other hand, is increased in conditions of iron excess induced by a deficiency of the ferroxidase genes (*Cp* and *Heph*) [21]. FtMt expression is pervasive throughout the brain, but the level of expression changes depending on the region [22]. Interestingly, the regional distribution of FtMt is not consistent with the distribution of iron in different areas of the brain. An example is deep cerebellar nuclei, which are particularly rich in iron but do not appear to express FtMt more than other cerebellar cells. The analysis of homogenates of the murine midbrain, obtained from animals with low, medium, and high amounts of iron in the brain, indicated that there was no difference in the quantities of FtMt among the different samples, so FtMt expression in these regions appears to be independent from relative iron content [22]. 

Since the expression of mitochondrial ferritin is not regulated by the classical iron-regulatory IRE/IRP machinery, it was hypothesized that it may depend on the number and activity of mitochondria. The distribution of FtMt in ependymal cells is an example that supports this hypothesis. Ependymal cells do not contain high concentrations of iron but contain a high number of mitochondria. The eyelashes of ependymal cells promote the circulation of cephalo-rachidian liquid inside the cerebral ventricles. The mitochondria of these cells are located in the apical region, near the base of the eyelashes, to provide the energy necessary for movement; these cells showed high levels of FtMt, with a distribution that overlapped the mitochondrial area [22].

The lack of a correlation between the concentration of cellular iron and the regulation of the FtMt expression levels could explain why FtMt has not been identified in the *substantia nigra pars compacta*, an iron-rich region with few mitochondria. In addition, the lack of expression in the liver and spleen, well-known iron-rich tissues, confirmed the hypothesis that the expression of FtMt is mostly unrelated to iron deposits. It was pointed out instead that FtMt expression is restricted to specific cell types with the oxidative metabolic activity of the cell; this suggests a potential role played by FtMt protecting mitochondria from iron-induced oxidative damage [23]. From the overall results, it was possible to associate the expression of FtMt with high metabolic activity and oxygen consumption by the cells. An exception was hepatocytes, which are metabolically very active and contain a large number of mitochondria but do not express FtMt. The expression of FtMt is not simply related to the number of mitochondria or high energy consumption; it seems instead that the presence of FtMt, in certain cell types, depends particularly on the urgency of the energy needs, such as during acute exhaustive exercise [24].

### Regulation of FtMt Expression

Several studies have been dedicated to the role of FtMt, but there are few data to explain how its expression is regulated and which types of mechanisms tune the specificity of this protein’s expression in tissues. The *FTMT* does not contain introns and does not have the typical TATA box or CCAAT box upstream of the first ATG codon. The region of DNA where *FTMT* is located is highly methylated in healthy somatic tissues. In silico analyses identified the consensus-promoter sequences preserved in different species. The highest sequence identities were found in the mouse (70% identity) and in the macaque (93% identity) for the region on the human sequence from −2040 to +600. Transfection experiments in HeLa cells led to the identification of one region containing activating regulatory elements and a second one containing inhibitory regulatory elements. In addition, a minimum promoter region was also identified in the upstream 500 bp [25]. The transcription factors CREB, YY1, and SP1 were proven to be involved as activators while, the factors FoxA1, C/EBPβ and GATA2 were found to be linked to the inhibitory elements [25]. The activating factor YY1 (Yin Yang 1) is a multifunctional ubiquitous transcription factor that can induce an open chromatin conformation, allowing access to other transcription factors. In addition, YY1 can bind the C-AMP response element binding protein (CREB) factor, another activator of the *FTMT*. Under conditions of hypoxia and oxidative stress in neurons, the CREB factor is activated by phosphorylation induced by the enzyme CREB-kinase and assembles into a transcriptional complex capable of promoting histone acetylation by altering the conformation of chromatin [26]. The specificity protein 1 (SP1) factor is essential for cell growth and differentiation. Oxidative stress induces the formation of DNA-SP1 binding complexes as a response to cellular stress; neurons, in particular, exhibited this mechanism [27]. It is assumed that under oxidative stress conditions, YY1 binds to DNA by inducing a relaxation of chromatin that allows the subsequent binding of CREB and SP1. These three factors interact with the transcription machinery, allowing the expression of *FTMT*. Thus, the limited expression of FtMt to specific cellular types with high energy consumption and susceptibility to the formation of reactive oxygen species can be explained by its regulation by CREB and SP1. GATA2 (GATA binding protein 2), FoxA1 (Forkhead-box protein A1), and C/EBP β (CCAAT-enhancer binding protein β) were selected from DNA regions containing inhibitory elements as inhibitory transcription factors. All these factors play a very important role in the development and cell differentiation. It is very likely that these players act as inhibitors of *FTMT* during tissue differentiation processes, when FtMt expression is not required [25]. The GATA2 factor plays a key role in transcriptional regulation of erythropoiesis and is highly expressed in pluripotent hematopoietic stem cells and newly formed erythroid cells. Under physiological conditions, FtMt is not found in erythroid cells, but it was shown that, under pathological conditions (like the myelodysplastic syndrome refractory anemia with ring sideroblasts, MDS-RARS), it is highly expressed in sideroblasts [20]. Subjects carrying mutations in the GATA2 gene have a high risk of developing the myelodysplastic syndrome and this pathogenic event may be responsible for the expression of FtMt in mutated cells. In addition, the expression of FtMt in these pathological cells interferes with the JAK2/STAT5 signal path leading to ineffective erythropoiesis [28,29].

In silico analysis on the UCSC Genome Browser revealed the presence of an extensive CpG island in the promoter of the *FTMT* that indicates a potential epigenetic regulation as confirmed in the work of Guaraldo [25]. Tightly regulated levels of FtMt are essential to maintain the cellular homeostasis of iron, and epigenetic regulation has been shown to play a role in maintaining and regulating FtMt levels in different tissues. Methylated cytosines in the CpG islands of the *FTMT* gene are probably recognized by the methyl binding protein MeCP2, which in turn recruits histone deacetylases that keep chromatin in a condensed state and prevent gene expression. Surely, further studies are needed to better understand the mechanisms that regulate the expression of FtMt, but it is already possible to hypothesize a potential application in therapy. The findings so far indicate that induction of FtMt expression through epigenetic therapy could be useful in diseases characterized by high oxidative stress.

Hypoxia is another condition that can upregulate FtMt expression [30]; functional binding sites for hypoxia-inducible factor 1α (HIF-1α) were identified in the promoter region of human *FTMT*. The induction of protein and its ability to prevent hypoxia-induced tissue damage was demonstrated in both in vitro and in vivo models [30]. 

## 3. Physiological Role of Mitochondrial Ferritin 

Studies done on different cellular models have proposed that FtMt plays a role in reducing toxic iron excess under conditions of oxidative stress [31,32]. FtMt expression in HeLa cells protected the mitochondria from oxidative damage induced by H_2_O_2_, antimycin A, or prolonged growth in a glucose-free medium (which stimulates mitochondrial respiration). Most of the effects caused by oxidative damage, such as the release of cytochrome c and the inhibition of mitochondrial ISC enzymes, were abolished by the expression of FtMt [17]. These results indicate that FtMt protects mitochondria by regulating local iron availability, thus making cells more resistant to oxidative damage [18]. The control of the formation of reactive oxygen species (ROS) through the regulation of mitochondrial iron availability, resulting in a cytoprotective effect, seems to be the main function of FtMt. Thus, FtMt expression may be important in cells where mitochondria are particularly exposed to ROS specially those mitochondria with excessive iron content. Under physiological conditions, mitochondrial ferritin is expressed at very low amounts mainly in cells with high metabolic rates, with the exception of the testis, where it is highly expressed. Thus, the study of its physiological role has been mainly carried out through overexpression experiment in yeast [31], cell lines [17,32,33], and flies [14]. All these studies were in agreement in demonstrating that overexpression of FtMt can induce a relocation of iron from cytosol to mitochondria, making it unavailable for metabolic functions. In particular, the analysis of HeLa cell clones expressing FtMt indicated an interesting link between mitochondrial ferritin and oxidative phosphorylation [34]. When the FtMt was expressed, the availability of iron for the synthesis of mitochondrial enzymes was reduced. Iron was seized within the FtMt and the synthesis of ISC was inhibited. In addition, it was observed that under conditions of increased cellular respiration, the amounts of FtMt were reduced, probably because of increased degradation of the protein [34]. In fact, in a medium with galactose, cells showed a much smaller amount of FtMt than cells grown in a medium with glucose. Thus, FtMt regulates the pool of labile iron in mitochondria and ROS productio, by protecting the cell from oxidative stress (Figure 2). On the other hand, the intensity of cellular respiration regulates the availability of iron for the synthesis of ISC-enzymes and heme by controlling the degradation of FtMt (Figure 2). This mechanism confirms that the FtMt expression is important in cells with high respiratory and metabolic activity, where the protein has a protective role.

In general, its protective role is exerted in all those situations where it is necessary to control iron excess, as in the case of erastin-induced ferroptosis, where the upregulation of FtMt becomes advantageous [35].

Unexpectedly, it was also observed that high amounts of FtMt increase damage due to oxidative stress through an iron-mediated mechanism [33]. The sensitivity of the cells to oxidative stress was due to cytosolic iron deficiency and iron subtraction from cytosolic ferritin [33]. High amounts of FtMt prevented the recovery of basal amounts of cytosolic ferritin. Since cytosolic ferritin plays a cytoprotective role under oxidative stress conditions, cells overexpressing FtMt were more susceptible to oxidative damage induced by the ferrous ion. Iron was released from cytosolic ferritin and mobilized to other compartments where it triggered the Fenton reaction causing severe damage to the cell. An even more important observation is that the lysosomes of cells overexpressing FtMt are less stable than those of control cells. Iron-rich mitochondria, because of high amounts of FtMt, can undergo mitophagy, increasing the concentration of Fe^2+^ in lysosomes. The formation of free radicals destabilizes the lysosomal membrane by making these organelles sensitive to oxidative stress. This mechanism may partly explain why cells overexpressing mitochondrial ferritin are more sensitive to oxidative stress [36]. Recent work described the induction of FtMt by the iron chelator deferiprone, resulting in increased mitophagy [37]; however, the results were affected by the use of a commercial polyclonal anti-FtMt peptide antibody, for which there were no available data about the lack of cross-reactivity with other proteins, in particular with cytosolic ferritin. Thus, more experiments are necessary to confirm these data.

From the overall in vitro studies, FtMt seems to have a dual effect: Beneficial and detrimental, depending on its amount and specific cell function. In vivo studies were performed prevalently on mouse models. *Ftmt*-deficient mice did not show any particular phenotype and were healthy [38]. After this first evidence, other *Ftmt* transgenic and null mice were developed [39,40]. None of them showed any noticeable effects of the increase level or absence of FtMt under physiological conditions, demonstrating that the gene is not essential for life. The only effect was the reduced male fertility of *Ftmt*−/− mice [41]. However, these animal models became informative when they were analyzed under oxidative conditions where FtMt expression becomes important and has a protective role. This FtMt defensive effect was detected on *Ftmt*−/− mice treated with several neurotoxic and cardiotoxic agents, such as 6-hydroxymomine (6-OHDA) [42], β-Amyloid [43,44] and doxorubicin [40]. The absence of FtMt in all these cases of cellular stress promoted cell damage and death. 

## 4. Role of Mitochondrial Ferritin in Pathological Condition 

Cytosolic ferritin with alterations of the *FTH* [45] and *FTL* [46,47] genes is characterized by dominant phenotypic transmission and causes disorders with a spectrum of phenotype severities depending on the site where the mutation occurred (for a comprehensive review see [48]); while genetic alteration in *FTMT* seems not to cause diseases. Mutations in the *FTMT* were first identified by a DHPLC-based screening as one polymorphism (P45H) and rare heterogeneous missense variations, but were not statistically significant when associated to diseases [49]. A mutation in the FtMt coding gene was then identified in patients affected by age-related macular degeneration (AMD), a multifactorial disease affecting the central retina (macula). The patient was described as heterozygous for two genetic changes: A complex deletion/insertion (c.437_450delinsCT) in *FTMT* and a missense p.Leu973Phe (c.2919G > T) mutation in exon 20 of *ABCA4*. The computational analysis of the two gene mutations and the experimental work on the FtMt recombinant variant showed a severe structural impairment for the FtMt variant and a mild destabilizing effect for ABCA4. These data suggested that a non-functional FtMt might be associated with AMD pathogenesis [50]. The inactivation of FtMt could lead to an increased sensitivity to oxidative stress because of iron favoring the development of the disease. 

In addition, FtMt also seems to play a role in pathological conditions characterized by defects in mitochondrial homeostasis and respiratory activity. In particular, its expression was altered in sideroblastic anemia, Friedreich ataxia, restless leg syndrome, and Alzheimer’s disease. A description of diseases and the involvement of mitochondrial ferritin is detailed in Table 2.

### 4.1. Mitochondrial Ferritin in Sideroblastic Anemia 

Sideroblastic anemia (SA, OMIM 300751) is a disease of hemoglobin synthesis, which is characterized by peculiar staining of erythroblasts. Perl’s staining of erythroblasts shows granules that are arranged in a ring around the nucleus (sideroblasts). The highlighted granules represent iron-overloaded mitochondria. This can occur in the X-linked sideroblastic anemia (XLSA, OMIM 301300) form or in sporadic forms, such as refractory anemia with ring sideroblasts (RARS) [20,29]. The XLSA form is characterized by missense mutations in the gene for the enzyme ALAS2 that alter its interaction with the substrate. The RARS form is more common than the X-bound form. Immunocytochemical analyses carried out on bone marrow smears of these patients revealed that FtMt is expressed at high levels in sideroblasts. In contrast, in healthy erythroblasts, FtMt was practically absent. Iron deposition in mitochondria appears to be due specifically to the presence of FtMt [29,51] indicating that FtMt is related and plays a role in the development of sideroblastic anemia [28,29]. 

Samples from patients with refractory anemia not characterized by the presence of sideroblasts had low positivity for FtMt [29,51]. Instead, samples obtained from patients with sideroblastic anemia expressed mitochondrial ferritin in many erythroblasts. In the case of X-linked sideroblastic anemia, 82 to 90% of erythroblasts expressed FtMt [29]. Mitochondrial ferritin was found at all stages of erythrocyte maturation [56], and in all cases, the characteristic ring arrangement of iron granules was evident. In contrast, FtH has widespread localization and is not found in all erythroblasts. In addition, in some patients with sideroblastic anemia, FtMt was also found in erythrocytes (called siderocytes). Based on the comparison between different forms of anemia it was concluded that mitochondrial ferritin is expressed exclusively in sideroblastic anemia [29]. The discovery that iron in sideroblast mitochondria is contained in FtMt, and that this protein is expressed exclusively in this condition, has profound clinical and pathophysiological implications. The development of techniques that specifically identify the FtMt would allow the early diagnosis of sideroblastic anemia. It would also make it possible to distinguish between the different forms of sideroblastic anemia, allowing an early prognosis on the development and possible risks of developing leukemia [51].

### 4.2. Role of Mitochondrial Ferritin in Friedreich’s Ataxia

Friedreich ataxia (FRDA, OMIM #229300) is an autosomal recessive neurodegenerative disease associated with cardiomyopathy. It is caused by polymorphic expansion of the GAA triplet in the frataxin gene, which alters its expression, leading to progressive damage to the nervous system over time. It is the most common form of hereditary ataxia [57]. Frataxin is a mitochondrial protein with iron-chaperone function for the synthesis of ISC, and its altered expression leads to mitochondrial iron overload and the production of ROS [58,59]. FtMt was identified in the mitochondrial fraction of cardiomyocytes and fibroblasts obtained from two FRDA patients [52] and from an autopsy sample of FRDA hearts [53]. This discovery suggested that mitochondrial ferritin expression could be a response to this pathology. In Friedreich ataxia, the cells are under chronic oxidative stress and, in this case, an increase in FtMt expression would be beneficial for the cell. In fact, the overexpression of mitochondrial ferritin in FRDA fibroblasts decreased ROS formation and maintained the activity of ISC-enzymes [34]. As also demonstrated in yeast and HeLa cells, FtMt expression can partly compensate for frataxin deficiency since these two proteins have some common properties [31,33,60]. Even if further studies are needed to verify whether FtMt can completely replace the function of frataxin, it has been hypothesized that FtMt decreases levels of ROS by seizing mitochondrial labile iron (mitLIP), reducing the detrimental phenotypic effects of FRDA, at least in in vitro models. Considering that epigenetic factors play a role in the control of *FTMT* expression and that the conditions to upregulate the expression of FtMt are milder [25] than those used to increase the expression of frataxin mRNA in the FRDA model [61], a therapeutic application of FtMt to prevent and reduce the effects of Friedreich ataxia has been proposed. 

### 4.3. Role of Mitochondrial Ferritin in Restless Legs Syndrome (RLS)

Restless Legs Syndrome (RLS, OMIM 102300) is a neurological disease found in 5–10% of the population [62] and is more common in women. Patients report a feeling of discomfort in one or both legs (involving the arms in the most severe cases) with the irresistible need to move them. This feeling of restlessness improves if the patient moves or massages the legs. The disorder often occurs in the evening following circadian rhythms and can also occur at night during sleep [63]. Magnetic resonance imaging (MRI) and transcranial sonography (TCS) of RLS patients have indicated a decreased amount of iron in the *substantia nigra* [64]. In addition, cerebrospinal fluid analysis has shown decreased amounts of ferritin and increased amounts of transferrin, indicating a state of iron deficiency. Immuno-histological studies have confirmed these data, particularly in neurons containing neuromelanin located in the *substantia nigra* [65]. All these results indicated that in RLS there is an alteration of cellular iron homeostasis, and a possible explanation for this iron deficiency in neurons could be the higher expression of FtMt. FtMt levels, in particular, were shown to be significantly increased in the *substantia nigra* of RLS samples compared to healthy controls [54], where FtMt is mainly present in neurons containing neuromelanin. There is evidence that the amounts of cytochrome and oxidase were also incremented, indicating an increase in the number of mitochondria in RLS samples. The increase in FtMt along with the increase in the number of mitochondria explain the decrease in cytosolic iron in the cells of RLS patients. As a result of this process, the cell experiences iron deficiency and the expression of cytosolic ferritin decreases. Iron deficiency in the cytosol, from iron relocation to the mitochondria by the FtMt, reduces the production and maturation of iron-dependent proteins. In fact, RLS cells express lower levels of the regulatory protein IRP1 and other ISC-proteins. Iron is also removed from the cytosol to maintain the high number of mitochondria, indicating the high metabolic activity of neural cells in RLS. The activation of mitochondrion genesis in this pathology leads to altered metabolism of iron and an inability to manage its daily physiological variations. Although the role of FtMt in even this pathology is not completely clear, its increase seems to favor the destabilization of iron homeostasis. Unlike Friedreich ataxia, in RLS, high amounts of mitochondrial ferritin appear to be harmful to cells.

### 4.4. Role of Mitochondrial Ferritin in Neurodegenerative Diseases

Iron overload seems to stimulate an auto-toxic circle that results in neurodegeneration. Indeed, an increase in iron concentrations is also observed in several neurodegenerative diseases, including the most frequent ones such as Alzheimer’s disease (AD) and Parkinson’s disease (PD) [66]. The cause of Alzheimer’s appears to be related to the alteration of the metabolism of the amyloid precursor protein (APP), leading to the formation of a neurotoxic substance: β-amyloid (Aβ) [67]. Experimental results have suggested that there is an interaction between iron metabolism and the metabolism of the β-amyloid protein [68,69]. In-situ hybridization, FtMt was shown to be expressed at much higher levels in the temporal cortex neurons of AD patients than those of healthy subjects [55]. These data support a possible involvement of FtMt in the pathogeneses of AD. Experiments in IMR-32 neuroblastoma cell lines have shown that, under oxidative stress conditions induced by the addition of the fragment Aβ_25–35_, FtMt levels increase, reducing oxidative damage through Erk/Pt 38 kinase signaling [70]. Instead, the knockdown of the *Ftmt* in mice greatly increases the neurotoxic effect of fragment Aβ_25–35_, increasing oxidative stress and apoptosis [70]. The data available so far indicate that FtMt expression increases in the brains of Alzheimer’s patients and that it has a neuroprotective role reducing the toxic effects of β-amyloid.

PD also has many characteristics associated with mitochondrial dysfunction and alterations in iron homeostasis. These occur when dopamine production in the brain drops substantially because of the degeneration of neurons of the *substantia nigra*. The amounts of iron present in the *substantia nigra* and *globus pallidus* are higher than in other areas of the brain, and in PD they are further increased. Iron accumulates particularly in Lewy bodies, one of the characteristic signs of PD [71]. In addition, iron promotes the aggregation of the α-synuclein protein, which is one of the main components of Lewy bodies [72].

To study the effect of FtMt expression in PD, a cellular model was obtained in the SH-SY5Y neuroblastoma line [42]. The presence of FtMt prevented the alteration of iron redistribution caused by 6-hydroxidopamine, leading to iron deficiency in the cytosol of SH-SY5Y cells. Reduced development of ROS, lower lipid peroxidation, and the maintenance of mitochondrial membrane potential were also observed. The overexpression of FtMt also increased the levels of the antiapoptotic protein Bcl-2, saving neuronal cells from apoptosis. Thus, the studies carried out so far have indicated that increased expression of FtMt in both PD and AD protects neuronal cells and inhibits the toxic effects of oxidative stress.

### 4.5. Role of Mitochondrial Ferritin in Ischemic Stroke

Deregulation of FtMt leads to an even worse prognosis in the case of ischemic stroke. Ischemic stroke is one of the most common brain diseases, but the molecular mechanisms behind this condition are not yet clear. Some evidence indicates that an excess of iron is a risk factor in the development of ischemia and that the iron amount is enhanced in ischemic brains [73,74]. Reperfusion is also a strong enhancer of ROS development [75]. Very recently, a study reported that mice with cerebral ischemia/reperfusion (I/R) also have increased *Ftmt* expression in the brain [76]. *Ftmt*−/− mice suffered more severe cerebral damage and neurological deficits accompanied by the molecular characteristics typical of ferroptosis. Some of these were increased lipid peroxidation and altered levels of glutathione (GSH). The ablation of *Ftmt* promoted ischemic inflammation and the reduction of ferroportin-1. Under these conditions, free iron levels greatly increased, favoring ferroptosis in the brain. In contrast, if the overexpression of FtMt was induced, these effects were inhibited. From the results, it is clear that FtMt plays a very important role in protecting against brain damage due to ischemic stroke [76]. With this evidence, it is also possible to suggest that FtMt might be considered as a new therapeutic target to prevent and reduce the damage of this very frequent pathological condition. 

### 4.6. Mitochondrial Ferritin as a Tumor Growth Inhibitor

To maintain a high rate of proliferation, cancer cells require high amounts of iron to synthesize iron-dependent proteins and enzymes. The dependence of cancer cells on the availability of iron makes these cells sensitive to its deficiency. This sensitivity has stimulated research and it is known from several clinical studies that iron chelators are effective anti-neoplastic agents [77].

Knowing the role of FtMt in triggering a cellular iron deficient phenotype, it was proposed that its expression could inhibit tumor growth by removing iron from neoplastic cells. In particular, overexpression of FtMt in in vivo cancer cells also reduced the ability of neoplastic cells to proliferate [78]. The presence of mitochondrial ferritin has drastically changed iron and heme levels into xenograft tumors. FtMt-expressing tumors had high levels of non-heme iron but showed a phenotype consistent with iron deficiency in the cytosol. Transmission electron microscope cancer analysis revealed that mitochondria are overloaded with iron, showing a morphology very similar to that observed in the erythroblasts of patients with sideroblastic anemia [78]. In addition, it was demonstrated that FtMt affected the cell cycle, causing G1/S arrest, and upregulated the expression of tumor suppressors, proving it can inhibit neuronal tumor cell proliferation [79]. These data demonstrated the effectiveness of iron removal from cancer cells as a strategy to limit tumor growth.

## 5. Conclusions

Mitochondria need iron for the synthesis of the heme group and iron-sulfur clusters, but it also generates large amounts of H_2_O_2_ as a side product of cellular respiration. The interaction between iron and H_2_O_2_ can cause damage to the cell, and the damage is particularly dangerous in cells that have high respiratory activity. Therefore, it was not surprising to identify in the mitochondrion a protein with the characteristics of cytosolic ferritin that has adapted to a different milieu. However, the great difference in expression levels of the two ferritins, due probably to the difference between the two environments where the cytosolic and mitochondrial ferritin operate, limited the experimental approaches to exploring the physiological role of FtMt. It is necessary to pay close attention to the properties of commercial polyclonal antibodies not characterized by specificity. They do not distinguish the minor amino acid differences between cytosolic and mitochondrial ferritin. Considering the enormous difference in expression of the two ferritins, it is very likely that the signal obtained with this type of antibody is the result of cross-reactivity with cytosolic ferritin. Nevertheless, the dual role of this protein—positive and harmful—was underlined by several in vitro and in vivo studies. These studies highlighted indications that FtMt could be used both as a diagnostic and prognostic marker and as a target for developing new therapies. It remains to be clarified whether the lack of FtMt is the primary cause of a disorder and whether the modulation of its expression, which is useful as an inhibitor of neoplastic growth in stroke and in neurodegenerative disease, can also be considered a valid therapeutic approach in humans. Although an application of mitochondrial ferritin in therapy is still only a hypothesis, FtMt can be used as a prognostic and diagnostic biomarker in various situations. An example is the presence of FtMt in erythroblasts, which is a hallmark of sideroblastic anemia. In addition, the increased expression of FtMt in Alzheimer’s diseases can be a way to diagnose the disease early and to predict its progress.

## Figures and Tables

**Figure 1 cells-10-01969-f001:**
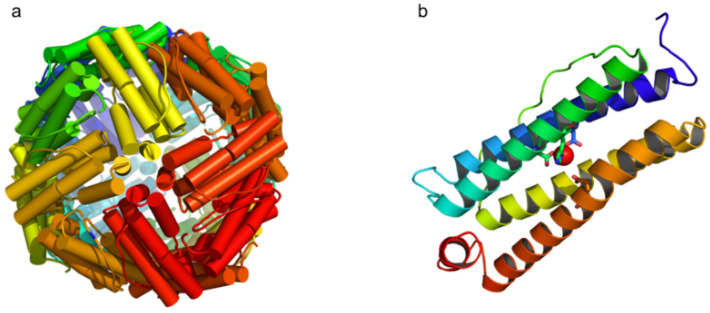
Scheme of the mitochondrial ferritin structure. (**a**) the three-dimensional structure of homopolymer mitochondrial ferritin and (**b**) ribbon representation of four-helix bundle subunit consisting of two couples of anti-parallel α-helices connected by a long loop and a short C-terminal α-helix. The residues involved in iron coordination are evidenced. The structures were obtained using PyMOL from Protein Data Bank (PDBsum 1r03).

**Figure 2 cells-10-01969-f002:**
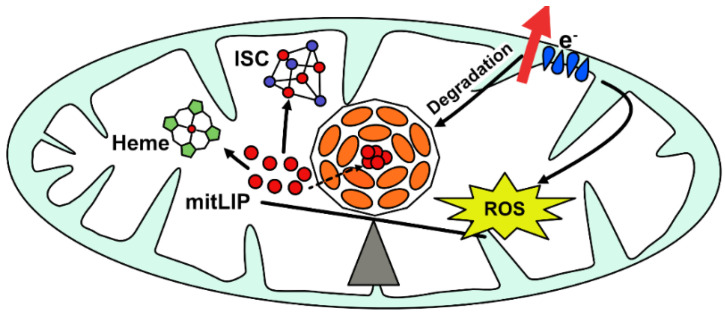
Scheme of the functional role of mitochondrial ferritin. Mitochondrial ferritin (FtMt) balances the amount of redox-active iron (mitLIP), necessary for the maintenance of Iron-Sulphur Cluster (ISC) and heme synthesis and the detrimental development of the ROS. Its greed for incorporating iron is counterbalanced by respiratory chain (e^−^) activity, which would appear to be involved in regulating FtMt degradation. In physiological conditions, FtMt expression is detectable only in cells characterized by intense respiratory activity, while in pathological conditions, it is associated with iron overloads in the mitochondrion and iron deprivation in the cytosol.

**Table 1 cells-10-01969-t001:** The major differences among cytosolic and mitochondrial ferritins.

	H Chain (FtH)	L Chain (FtL)	Mitochondrial Ferritin (FtMt)
Chromosome	11	19	5
Introns	3	3	None
Length of the encoded protein	175 amino acids	183 amino acids	242 amino acids
NH_2_ processing	N-terminal methionine cleavage, N-terminal block	N-terminal methionine cleavage, N-terminal block	N-terminal leader sequence cleavage
Structure of ferritin	Heteropolymers H/L	Heteropolymers H/L	Homopolymers
Catalytic site	Ferroxidase center	No ferroxidase center (facilitates iron nucleation)	Ferroxidase center
Fe-dependent regulation	IRE-dependent post-transcriptional regulation	IRE-dependent post-transcriptional regulation	No
Transcriptional regulation *	Dependent on the ARE element	Dependent on the ARE element	Dependent on the HRE element
Tissue expression	Ubiquitous	Ubiquitous	Cellular specific

* ARE = Antioxidant Response Element, HRE = Hypoxia Response Element.

**Table 2 cells-10-01969-t002:** Diseases involving FtMt.

Disease	FtMt Gene	FtMt Protein	FtMt Expression	Detected in *	Reference
Macular Degeneration (MD)	Mutated	Not Functional?	--	--	[50]
Sideroblastic Anemia (SD)	Wild Type	Normal	Upregulated	Sideroblasts	[20,51]
Friederich’s Ataxia (FA)	Wild Type	Normal	Upregulated	Cardiomyocytes, Fibroblasts, Heart	[52,53]
Restless Legs Syndrome (RLS)	Wild Type	Normal	Upregulated	*Substantia Nigra*	[54]
Alzheimer’s Disease (AD)	Wild Type	Normal	Upregulated	Temporal Cortex Neurons	[55]

* The presence of FtMt was investigated in the tissues and cell types reported.

## Data Availability

Not applicable.

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
