# Peer review of "Mitochondrial Ferritin: Its Role in Physiological and Pathological Conditions"

_cells, 2021, doi:10.3390/cells10081969_

Round 1

Reviewer 1 Report

Understanding the full function of FtMt is still need to be investigated, although FTMT knockout mice does not show any obvious phenotype. In this review, the authors, first, described an overview of FtMt structure, function and regulation of its expression. Next, the authors described the role of FtMt in physiology, mostly based on in vitro studies, and in pathologies like sideroblastic anaemia, Friedreich ataxia, RLS, Alzheimer's disease, ischemic stroke and cancer. Information described here provides valuable insights for future studies of FtMt functions.

Major concern:

The second part of the review (physiological role of mitochondrial ferritin) described the MtMt functions identified on cell models. Some of these functions were already discussed in the first part (Page 3, lines 98-109). Thus, the second part of the review could be included in the first part, (FtMt: structural, biochemical, functional and regulatory peculiarities). The inverse could also make the review easier to read: moving the function of FtMt discussed in the first part to the second part of the review.  

Minors concerns:

  • Within the text on the page 3, lines 98-108 or within the text on the page 6, line 258, the authors should discuss this recent study. EMBO Rep (2020)21:e50202https://doi.org/10.15252/embr.202050202

  • Page 2, lines 64-69. The authors discuss the expression level of FtMt in different tissues. These observations are more relevant within the text describing the expression of FtMt in page 3, lines 110-117.

  • Page 4, change “FtMt gene” by FTMT or FTMT.

  • It’s not clear why in the page 3, line 65, the authors indicate that : “FtMt coding transcript is present at high levels in the testis” (physiological condition) and in page 5, line 217, the authors indicate that : FtMt is expressed at very low level in physiological condition. Please clarify this point.

  • Figure legend is needed or at least include the details of the abbreviations within the figure

  • Page 6, line 258: Is not clear why the authors talk about cells :“When cells are exposed”, in animal model.

Author Response

We thank Reviewer 1 for the positive comments and suggestions that help us to improve the manuscript.

The manuscript has been fully revised, and the points raised by all the reviewers have been accepted, and the text modified accordingly. Following the request of all the reviewers some paragraph are slightly modified, thus the lane number of the entire manuscript is changed. The answers to the reviewers are referred to the new page number. The manuscript has been edited by the MPDI English editing service.

The answers to the reviewer 1 comments are detailed below.

Major concern:

Following reviewer suggestion we moved the function of FtMt discussed in the first part to the second part of the review.

Minor concerns:

  • On page 7 a discussion of the paper EMBO Rep 2020 has been inserted. “A recent work described the induction of FtMt by the iron chelator deferiprone, resulting in increased mitophagy [37] however, the results were affected by the use of a commercial polyclonal anti-FtMt peptide antibody, for which there were no available data about the lack of cross-reactivity with other proteins, in particular with cytosolic ferritin. Thus, more experiments are necessary to confirm these data.”
  • As suggested, the observations about the expression level of FtMt in different tissues has been moved on page 4.
  • We checked all the FTMT and changed into italics.
  • Thank you for the observation, we clarified the sentence previously on page 5 lane 217 with the following sentence on page 6:

“Under physiological conditions, mitochondrial ferritin is expressed at very low amounts mainly in cells with high metabolic rates, with the exception of testis, where it is highly expressed”

  • Figure legend was already present in the submitted manuscript but for unknown reason was incorporated into the main text. Now the figure legends are highlighted in yellow with the character size of 9 points.
  • We changed the sentence on page 6 lane 258 with the following:

“However, these animal models became informative when they were analyzed under oxidative conditions where FtMt expression becomes important and has a protective role.” Now on page 7.

Reviewer 2 Report

The review provides a comprehensive and clear summary of current knowledge on the role of mitochondrial ferritins in health and disease.  The manuscript is well-written and highlights important advances in the field of mitochondrial iron metabolism.

I have few minor comments:

1) It would be beneficial to provide the figure legend for the scheme of the functional role of mitochondrial ferritin in figure 1.

2) lane 117. “ In addition, FtMt has not been  found in epithelial cells with a mechanical barrier function.” Since the authors mentioned that FtMt is expressed only in specific tissues, the accent on only epithelial cells seems not clear.

3) Manuscripts gives a comprehensive overview of new development in the field. Authors might consider discussing a recent publication (2020) on the role of mitochondrial ferritin in iron-deprivation-induced mitophagy (PMID: 32975364).

Author Response

We thank Reviewer 2 for the positive comments and suggestions that help us to improve the manuscript.

The manuscript has been fully revised, and the points raised by all the reviewers have been accepted, and the text modified accordingly. Following the request of all the reviewers some paragraph are slightly modified, thus the lane number of the entire manuscript is changed. The answers to the reviewers are referred to the new page number. The manuscript has been edited by the MPDI English editing service.

The answers to the reviewer 2 comments are detailed below.

  • Figure legend was already present in the submitted manuscript but for unknown reason was incorporated into the main text. Now the figure legends are highlighted in yellow with the character size of 9 points.
  • Thank you for the observation, the unnecessary sentence has been removed in the new version.
  • On page 7 a discussion of the paper EMBO Rep 2020 has been inserted. “A recent work described the induction of FtMt by the iron chelator deferiprone, resulting in increased mitophagy [37] however, the results were affected by the use of a commercial polyclonal anti-FtMt peptide antibody, for which there were no available data about the lack of cross-reactivity with other proteins, in particular with cytosolic ferritin. Thus, more experiments are necessary to confirm these data.”

Reviewer 3 Report

In the review "Mitochondrial ferritin: its role in physiological and pathological condition" (cells-1309408) by Levi et al, the current knowledge of mitochondrial ferritin is described. The authors have made a lot of effort to include as much published papers as possible, however they could improve the comprehensiveness and the philologic outline of the manuscript. Introduction of more illustrative figures would also be in place. In addition, the manuscript contains quite a lot of typos (of which only a few are commented below) and errors in grammar. English language help should be used to improve it. The manuscript in its present state is not recommended for publication in Cells.

Major comments

The Abstract should be re-written to reflect/summarize the contents of the manuscript better.

The Introduction lacks more general description how iron is taken up, transported, imported into cells and imported into mitochondria. Does free iron exist in cells or is it always bound to proteins or other chelators? What are the physiological concentrations of iron? This is important for the benefit of the general reader.

Lines 80-84. The structure of ferritin: it would be in place with a figure or a better description of the domain fold. For example are the five alpha-helices packed in parallel or antiparallel bundle? Where in the structure is the iron binding site?

Section 3. A table should be in place with the diseases described and their association with FtMt levels.

Minor comments

Line 3 (in the title): "condition" should be "conditions".

Line 40: "are expressed in the same compartments". What do the authors want to say with this statement. Is not also the FtMt "expressed" in the same compartments?

Lines 60-61: "lower homology to" maybe should be "less similar to" because then it is stated that they may have "a different phylogenetic origin" i.e. they are not homologues.

Line 77: "cito" should be "cyto"

Line 80: "E. Coli" should be spelled out and in italics.

Line 83: "helixes" should be "helices".

Line 110: "hystochemical" should be "histochemical".

Line 116: "highest positivity". It is not clear what the authors mean.

Table 1 should be displayed early on in the manuscript because some of its contents appear in the text early on.

Lines 350-351: "5%-10% of the population". These figures seem a bit exaggerated and should be referenced to scientific literature.

Author Response

We thank Reviewer 3 for the positive comments and suggestions that help us to improve the manuscript.

The manuscript has been fully revised, and the points raised by all the reviewers have been accepted, and the text modified accordingly. Following the request of all the reviewers some paragraph are slightly modified, thus the lane number of the entire manuscript is changed. The answers to the reviewers are referred to the new page number. The manuscript has been edited by the MPDI English editing service.

The answers to the reviewer 3 comments are detailed below.

Reviewer #3:

Major comments:

The abstract has been revised adding more details on the content of the review.

The editors of the special issue on “Mitochondria meet iron metabolism: facts and opportunities” ask us to write a review on FtMt, thus we inserted only a short general description on cellular iron metabolism at the beginning of the introduction paragraph, considering that a deeper description is a topic of other reviews in this special issue.

A figure with the FtMt structure has been inserted with the requested description in the figure legend.

A table with the summary of the diseases associated to FtMt have been added.

Minor comments:

  • Title has been corrected.
  • The sentence originally in lane 40 has been rephrased as following “The H- and L-chains are expressed in the cytosol compartment and are under the same iron-dependent post-transcriptional control, but do not tend to assemble into homopolymer molecules.” Now on lanes 98-100.
  • The sentence originally in lanes 60-61 has been rephrased as following “Plants and Drosophila melanogaster possess the FTMT gene but it is less similar to mammals’ genes, as it is interspersed with introns indicating a different phylogenetic origin” now on page 3.
  • All the spelling errors have been corrected.
  • The sentence originally in lane 116 has been rephrased as following “Among germ cells, the ones showing the highest level of FtMt were mature sperm cells, where mitochondria respond very actively to the high energy needed for movement“ now on page 4.
  • As suggested Table 1 has been moved earlier in the manuscript.
  • We added the reference where we found the percentage of incidence of RLS in general population.

Round 2

Reviewer 1 Report

The new version is well structured and very clear.

You could (but not necessary) add a heading on the page 4, line 142 : Regulation of FtMt expression.